

# Review of snow phenology variation in the Northern Hemisphere and its relationship with climate and vegetation

Hui Guo[1], Xiaoyan Wang[1], Zecheng Guo[1], Gaofeng Zhu[1], Tao Che[2], Jian Wang[2,3], Xiaodong Huang[4], Chao Han[1], and Zhiqi OuYang[1]

[1] College of Earth and Environmental Sciences, Lanzhou University, Lanzhou 730000, China.
[2] Heihe Remote Sensing Experimental Research Station, Key Laboratory of Remote Sensing of Gansu Province, Northwest Institute of Eco-Environment and Resources, Chinese Academy of Sciences, Lanzhou 730000, China.
[3] Jiangsu Center for Collaborative Innovation in Geographical Information Resource Development and Application, Nanjing 210023, China.
[4] Key Laboratory of Grassland Agro-ecosystems, College of Pastoral Agriculture Science and Technology, Lanzhou University, Lanzhou 730000, China.

*Correspondence to*: Xiaoyan Wang (wangxiaoy@lzu.edu.cn)

**Abstract.** Snow phenology, recurrent seasonal patterns in snow cover and snowfall, has been significantly affected by global warming. Through the interaction with the climate, the dynamic variability of snow phenology affects the regional climate environment, vegetation ecosystem, soil properties, agricultural water resources, snow disasters and animal migration. First, this study compares the advantages, disadvantages and applicability of different sources of observation data and the principal research methods involved in studying snow phenology. Then, this work discusses the spatiotemporal variability and changing trends of snow phenology in the Northern Hemisphere, and summarizes the relationship between climate, vegetation and snow phenology. Finally, this review highlights the key areas related to snow phenology that require further study. Overall, during the past 50 years in the Northern Hemisphere, the snow cover end date (SCED) has shown a significantly advanced trend, the snow cover onset date (SCOD) has also been occurring slowly earlier, and the snow cover days (SCD) has shortened, but these two trends are not significant. The snow phenology variation is closely related to climate factors, atmospheric circulation, vegetation status and some spatial factors. Snow cover impacts climate change through interactions with atmospheric circulation systems. The rise in temperature will delay the SCOD, and the SCED is closely related to the temperature of the snowmelt season. The interaction between seasonal snow cover and climate will either stimulate or impede vegetation growth. With the change in snow cover, especially the decrease in snow cover in the melting stage can impact the climate change, the rise in temperature will change the growth conditions and extend the vegetation growth season. The relationship between snow cover and vegetation is inconsistent in different elevations and latitudes. Snow phenology variation is very complex and is the result of the combined action of many factors. Additionally, snow phenology will also have a great impact on the cryosphere. Therefore, we must understand snow phenology variation and prepare for future changes.



## 1 Introduction

The Sixth Assessment Report of the Intergovernmental Panel on Climate Change (IPCC) notes that human beings have accelerated climate warming at an unprecedented rate, which has accelerated the melting rate of snow (IPCC, 2021). 98% of the seasonal snow cover on global land is located in the Northern Hemisphere (Armstrong and Brodzik, 2001), which significantly affects Earth's climate change. The reduction in large scale snow areas will lead to widespread and rapid changes in the atmosphere, ocean, cryosphere and biosphere. Continuous changes in regional snow cover will affect periodic changes in the ecosystem, and extreme snow events may also have destructive social and economic effects (Harpold and Brooks, 2018).

Snow phenological parameters such as snow cover days (SCD), snow cover onset date (SCOD), and snow cover end date (SCED) are important indicators of climate change as well as providing input parameters for global energy balance, climate, hydrology and ecological models (Liston, 1999; Notarnicola, 2020; Liston and Hiemstra, 2011). These parameters are used to describe the response of snow cover variation to climate change and show a regular seasonal and interannual variation characteristics (Sun et al., 2020). Related to global warming, the rise in temperature delayed the SCOD in autumn, and the spring snow melt also occurred earlier. Earlier seasonal snowmelt may lead to an increase in snowmelt-related runoff and an earlier peak in snowmelt-related floods in early spring (Beebee and Manga, 2004). Climate change leads to variability in snow cover. In turn, changes in snow cover will inevitably affect climate and atmospheric circulation (Huang et al., 2021; Yu et al., 2013., Haynes et al., 2014; Arnell, 2005; Flanner et al., 2011). Specifically, snow cover anomalies will cause adjustments of the surface thermal conditions via the radiation effect, which alters heat conduction between the surface and the atmosphere. This then causes a change in temperature, resulting in abnormal circulation and precipitation (Chen et al., 2003). In addition, snow phenology variations have been shown to be related to vegetation activity (Grippa et al 2005, Peng et al 2010), and the interaction between seasonal snow cover and climate either stimulates or inhibits vegetation growth (Dong et al., 2013; Inouye, 2008; Wipf and Rixen, 2010). In general, snow phenology plays an important feedback role in climate change through its characteristics of high reflectivity and low thermal conductivity (Warren,1982). Furthermore, snow phenology affects terrestrial ecosystems by changing the air temperature, circulation patterns and photosynthesis of green vegetation (Gerland et al., 2000; Saunders, Qian, and Lloyd, 2003). Therefore, it is essential to study the characteristics of snow phenology variation, the response of snow phenology to climate and atmospheric circulation, and the relationship between snow phenology and vegetation ecosystems.

Although many studies have researched snow phenology variation and its relationship with climate and vegetation, there are still many uncertainties and limitations, such as inconsistent accuracies and variability in data obtained by different sensors. In this work, we summarize the data, methods, results, and the relationships that climate and vegetation have with snow cover in the study of snow phenology in the Northern Hemisphere. Section 2 summarizes the data and methods used to study snow phenology. Section 3 presents the temporal and spatial characteristics of snow phenology variation in the Northern



Hemisphere. Section 4 discusses the interaction between snow phenology variation and climatic factors, atmospheric circulation and vegetation. Section 5 briefly proposes further snow phenology research in the future.

## 2 Condition of snow phenology research technology

### 2.1 Data Sources

### 2.1.1 Ground observation data

The snow information obtained from ground-based observations has a long-term and rich historical record. These records include snow depth data from snow gauges in Russia, Canada, the United States and Europe that extend back to the 19th century. When measuring snow depth, a flat and open place near the observation station or some other representative position is generally preferred. Daily mean snow depth is the average snow depth of multiple measurements (CMA, 2017; ECA&D, 2013), and the measurement accuracy can be accurate to 0.1 cm according to the ground observation in China (Huang et al., 2020). With improvements in observation technology, snow depth can be monitored at many meteorological stations automatically and continuously by using ultrasonic snow depth detectors (Zhong et al., 2020; Huang et al., 2020; Vionnet et al., 2021; Michael et al., 2021). The time series of ground-based snow depth data (SD) is long, and it is often used to study the snow phenology variation or to verify the results of remote sensing analysis due to its high measurement accuracy. However, due to the heterogeneity of terrain and human activities, the station distribution is uneven, the observation range is limited, and the few ground stations are not enough to represent the actual snow cover distribution over a wide range, especially in alpine areas (Xiao et al., 2019; Wang and Xie, 2009; Wang et al., 2018a; Robinson et al., 1993). Table 1 summarizes the snow depth datasets available in the Northern Hemisphere. Note that the Canadian Historical Snow Water Equivalent (CanSWE) dataset replaces the Canadian Historical Snow Survey Data (CHSD). The previous dataset contained metadata errors and a large amount of duplicate data (Brown et al., 2019). The Global Historical Climatology Network-Daily (GHCN) dataset is the most comprehensive ground observation daily climate data worldwide, and includes records from 180 countries. Almost all snow depth ground observation stations are located in the Northern Hemisphere (Menne et al., 2012).

**Table 1. Ground observation snow depth datasets**

| Spatial Coverage | Temporal Coverage | Number of stations | Sources | Reference |
|---|---|---|---|---|
| China | 1951-2021 | 730 | China Meteorological Administration (CMA) http://data.cma.cn | Huang et al.,2020 |
| Europe | 1849-present | 187 | European Climate Assessment Dataset (ECAD) https://www.ecad.eu | Peng et al.,2013 |
| Russia | 1966- present | 599 | Russia Research Institute of Hydrographic Information-World | Zhong et al.,2021 |





| | | | Data Centre (RIHNI-WDC) http://www.wdcb.ru | |
| --- | --- | --- | --- | --- |
| Canada | 1928-2020 | 2607 | National Climate Data and Information Archive of Environment Canada https://open.canada.ca/data/en/dataset | Vionnet et al., 2021 |
| Global | 1880- present | 68832 | Global Historical Climatology Network (GHCN) https://www.ncdc.noaa.gov/cdo-web/datasets | Menne et al., 2012 |

### 2.1.2 Remote sensing data

Compared with ground observation data, remote sensing technology can detect snow information on a continuous scale.
Snow has a high reflectance in the visible band and a low reflectance in the shortwave infrared band. Based on this characteristic, the normalized difference snow index (NDSI) developed by optical remote sensing can effectively identify the snow cover extent (Hall et al.,1995; Salomonson and Appel, 2006; Hao et al., 2022). Moderate Resolution Imaging Spectroradiometer (MODIS) snow products, as the most common optical remote sensing snow products, have a high resolution and accuracy, but the data are limited by clouds and short time series. MOD10A1, MOD10A2 and MOD10C2
have also been widely used in the study of snow phenology in the Northern Hemisphere (see Table 2 for details). The MODIS snow cover algorithm and data from the Collection 6 products have been significantly improved, and the data content has increased compared with Collection 5 (C5), which was discontinued in 2016. Taking MOD10A1 as an example, the snow information is represented by NDSI, which replaces the fractional snow cover (FSC) in C5, and NDSI can describe the snow distribution more accurately than FSC (Riggs et al, 2016). The Advanced Very High Resolution Radiometer
(AVHRR) has provided a long time series at a high temporal resolution, but it requires a complex process to obtain the binary snow product.

Compared to optical remote sensing monitoring data, passive microwave data is not affected by clouds and have strong objective accuracy. This data mainly studies snow phenology by using an inversion algorithm to obtain snow depth or snow water equivalent (SWE). Global Snow Monitoring for Climate Research (Globsnow), which was released by the European
Space Agency in 1979, is considered the most influential daily global SWE dataset in the world, and its spatial resolution is 25 km (Huang, Li et al., 2019). However, due to the lack of mountainous data and serious data gaps from June to September, its wide application in the study of snow phenology variation is limited (Xiao et al., 2020). In addition, the most representative SWE data is from the Advanced Microwave Scanning Radiometer for the Earth Observing System (AMSR-E)/Aqua L3 Global Snow Water Equivalent EASE-Grids, which was available from 2002 to 2011. Based on the daily SWE,
the five-day maximum SWE and monthly average SWE granules are synthesized. The Advanced Microwave Scanning Radiometer2 (AMSR2) onboard the GCOM-W1 satellite was launched in 2012 as a follow up product of AMSR-E, and the daily SWE data can still be derived from AMSR2 L1R (Hu et al., 2016). Recently, the National Tibetan Plateau Data Center released a long-term record of daily snow depth over the Northern Hemisphere (NHSD) based on machine learning data, in which the determination coefficient increased to 0.81 from 0.23 compared with the ground observation data and the snow
depth products before fusion (Hu et al., 2021).



In addition to the above remote sensing data, there are cloud-free snow products for studying snow phenology variation. The Northern Hemisphere EASE Grid 2.0 Weekly Snow and Sea Ice Extent (NHSCE) generated by the National Oceanic and Atmospheric Administration (NOAA) began in 1966 and is the longest dataset including binary snow cover extent and sea ice data. NOAA began to develop an interactive multisensor ice and snow mapping system (IMS) widely used to obtain the large scale snow cover extent regardless of cloud coverage. The resolution of IMS products released in 1997 was 24 km. The data accuracy was continuously improved over time, and 4 km and 1 km products were released in 2004 and 2014, respectively (U.S. National Ice Center, 2008). The Near Real-Time SSM/I-SSMIS EASE-Grid Daily Global Ice Concentration and Snow Extent (NISE) is similar to the available IMS records and provides sea ice concentration and snow cover extent. The snow cover extent is derived from passive microwave satellite data, and its derivations use the brightness temperature measured by the satellite in the morning pass as inputs and rely on nearest neighbor interpolation (Brodzik and Stewart, 2016). The global daily snow depth analysis data from the Canadian Meteorological Center (CMC) is generated by in situ daily snow depth observations and is optimally interpolated by a first-guess field that is derived from a simple snow accumulation and melt model, which is based on analyzed temperatures and forecasted precipitation from the Canadian forecasts model. The dataset includes snow depth, monthly average snow depth and snow depth climatology for snow seasons. In addition, it also includes the snow water equivalent data (Brown and Brasnett, 2010).

**Table 2 Remote sensing snow cover datasets**

| Dataset | Spatial Coverage | Temporal Coverage | Spatial Resolution | Temporal Resolution | Data type | Sources |
|---|---|---|---|---|---|---|
| MOD10A1 | | | 500 m | 1 day | NDSI | |
| MOD10A2 | | | 500 m | 8 days | snow extent | |
| MOD10C1 | global | 2000-present | 0.05° | 1 day | NDSI | https://modis.gsfc.nasa.gov |
| MOD10C2 | | | 0.05° | 8 days | snow extent | |
| MOD10CM | | | 0.05° | 1 month | snow extent | |
| AVHRR | global | 1981-present | 0.05° | 1 day | reflectivity | https://www.ncdc.noaa.gov |
| GlobSnow | Northern Hemisphere | 1979-present | 0.25° | 1 day | SWE | https://nsidc.org/data/NSIDC-0595 |
| AMSR-E/ AMSR2 | global | 2002-2011/ 2012-present | 25 km | 1 day | SWE | https://modis.gsfc.nasa.gov |
| NHSD | Northern Hemisphere | 1980-2019 | 0.25° | 1 day | SD | http://data.tpdc.ac.cn/zh-hans |
| NHSCE | Northern Hemisphere | 1966-2020 | 25 km | 1 week | snow extent | https://nsidc.org |
| IMS | Northern Hemisphere | 1997-present | 24 km/4 km/1 km | 1 day | snow extent | https://nsidc.org |



| NISE | global | 1995-present | 25 km | 1 day | snow extent | https://nsidc.org |
|------|--------|--------------|-------|-------|-------------|-------------------|
| CMC | Northern Hemisphere | 1998-2020 | 24 km | 1 day | SD/SWE | https://nsidc.org |

### 2.1.3 Reanalysis data

The fifth generation of the European Reanalysis (ERA5), which is a comprehensive reanalysis dataset, can also be used to study snow phenology. It is the fifth generation of the global atmospheric reanalysis product provided by the European Center for Medium Range Weather Forecasts-ECMWF (https://www.ecmwf.int). ERA5 combines a large number of historical observations into global estimation using advanced models and data assimilation systems. Compared with ERA-Interim (1979-2019), ERA5 offers a longer time series (1950 to the present), an improved spatial resolution (from 79 km to 31 km), an improved temporal resolution (from monitoring intervals of 6 hours to one hour). The ERA5-land dataset contains snow information such as snow albedo, snow density, snow depth and snow water equivalent, and depicts land surface features at a higher spatial resolution (9 km) (Munoz et al., 2021). In addition, the Modern-Era Retrospective Analysis for Research and Applications, version 2 (MERRA2) is the latest generation of daily reanalysis data released by NASA (https://disc.gsfc.nasa.gov/datasets), which provides global average snow depth and snow cover fraction data with a spatial resolution of 0.5° from January 1980 to the present (Lin et al., 2019).

### 2.1.4 Simulation data

The Coupled Model Inter-comparison Project (CMIP) can forecast future snow cover variation. Various climate models can be used to predict the formation and melting time of snow under the conditions of different representative concentration pathways. The WCRP Working Group on Coupled Modeling (WGCM) launched the fifth coupled model inter-comparison project (CMIP5) in 2008. Compared with CMIP3, the models of this project have been improved to a certain extent, and the external forcing data used in the experiment are closer to the actual conditions (Xia and Wang, 2015). Therefore, the CMIP5 climate model can still better simulate the spatial distribution and the seasonal and interannual variation characteristics of snow cover in the Northern Hemisphere. However, CMIP5 underestimates the changing trend of snow cover extent in the Northern Hemisphere (Brutel et al., 2013), and the model performs poorly when simulating snow cover in areas of complex terrain, such as plateaus (Guo et al., 2013). The international coupled model inter-comparison project has now developed to the sixth stage (CMIP6) (https://esgf-node.llnl.gov/search/cmip6), and a new set of climate model experiments has been applied to predict snow cover variation (Zhou et al., 2019).

### 2.2 Extraction method of snow phenology parameters

Before determining the snow phenological parameters, it is first necessary to specify the start date of the snow year, and the researcher may use different starting and ending times in the research. Generally, the snow year starts in July, August or



early September and ends at the end of June, July and August of the following year (Notarnicola, 2020, Zhong et al., 2021;
Xu et al., 2017; Wang et al., 2017; Chen et al., 2015). There are different methods (hereinafter referred to as M1, M2, M3) to determine the SCOD and SCED (Table 3). In M1, combined with the local snow situation and considering the instantaneous snowfall, the days of continuous snow coverage are selected as one day or three days, though other values may also be selected (Xie et al., 2017; Zhong et al., 2021). M2 hypothesizes that the snow is stable during $D_1$ and $D_2$ and ignores instantaneous snowfall. M3 defines the SCOD and SCED according to the snow depletion curve. Snow cover days (SCD)
are defined as the total number of days with snow cover and snow cover duration days (SDD) refers to the number of days between SCOD and SCED within a snow year (Notarnicola, 2020; Saavedra et al., 2018).

**Table 3 The definition of snow phenological parameters**

| Snow phenological parameters | | Definition | References |
|---|---|---|---|
| SCOD | M1 | the first day on which the pixel is first covered with snow for at least some number of consecutive days | Notarnicola, C,2020 |
| | M2 | $SCOD = D_1 - SCD_1$ | Wang and Xie, 2009 |
| | M3 | the start date when the SCE exceeds 5% and remains above 5% for at least ten consecutive days | Dariane et al., 2017 |
| SCED | M1 | the last day on which the pixel is last covered with snow for at least some number of consecutive days | Notarnicola, C,2020 |
| | M2 | $SCED = D_2 + SCD_2$ | Wang and Xie, 2009 |
| | M3 | the end date when the SCE is less than 5% and remains below 5% for at least ten consecutive days | Dariane et al., 2017 |
| SCD | | $SCD = \sum_{i=1}^{n} Snow_i$ | Notarnicola, C,2020 |
| SDD | | $SDD = SCED - SCOD + 1$ | Sun et al., 2020 |

M1, M2, M3 are different definition methods; SCE is snow cover extent; $D_1$ and $D_2$ are fixed Julian days in the accumulation season and snowmelt season, respectively. $D_1$ and $D_2$ depend on the specific climate and may be around December and February, respectively. $SCD_1$
is the number of snow cover days before $D_1$ in the accumulation season, and $SCD_2$ is the number of snow cover days after snowmelt season $D_2$ (Wang and Xie, 2009). $Snow_i$ is snow pixel at a certain location.

## 3 Recent progress on snow phenology variation in the Northern Hemisphere

### 3.1 Snow phenology climatology

With abundant observation data and research methods, scholars have conducted some studies of snow phenology in the
Northern Hemisphere. The analysis of NHSCE snow cover data shows that the snow phenology has zonal and vertical distribution characteristics (Fig. 1), which is consistent with previous studies (Chen et al., 2015; Peng et al., 2013., Chen et



al., 2021). At high latitudes and altitudes SCOD occurs relatively earlier, SCED occurs later and SCD is longer compared to lower latitudes and elevations. Over the past 30 years, snow cover has been detected in September in northern Alaska, the Tibetan Plateau and the northern Himalayas. In the Arctic, the snow completely melted in June (Zhong et al., 2021). The

analysis results based on MODIS data show that at the middle latitudes in Eurasia, snow began to fall in November and melted in mid-February. With the increase in latitude, the SCOD is advanced to the end of September, and the SCED shifts into mid-April at 60 °N. In mountainous regions with high latitudes, the SCED occurs at the end of May, later than that in plain areas at the same latitude (Sun et al., 2020).

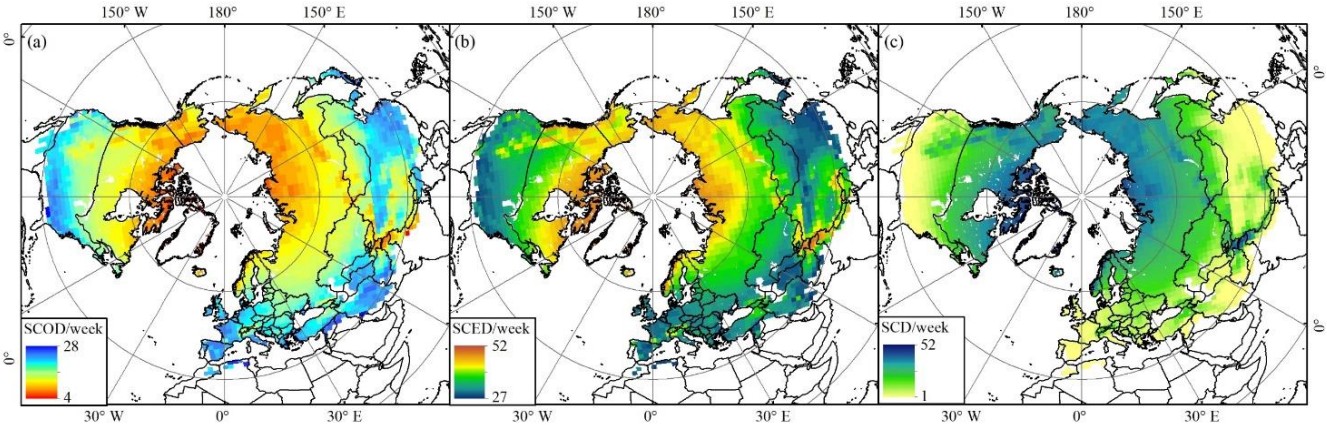

**Figure 1: Distribution of snow phenology over the Northern Hemisphere from 1972 to 2020 based on NHSCE snow cover data.**
**Averaged over 48-snow years (a) SCOD, (b) SCED, and (c) SCD (the study area excludes Greenland). The snow year is from 1 August to 31 July of the following year. SCOD lies between 1 August and 31 January, SCED lies between 1 February and 31 July, and SCD indicates the sum of snow cover weeks.**

The strongest spatial variability of the snow cover persistence period appears at 40°N-60°N. The SDD decreases
significantly with decreasing latitude, from approximately 30 weeks in the north to less than five weeks in the south (Zhang and Ma, 2018). The spatial distribution of SDD is relatively stable in low and high latitudes. In the Arctic, the number of snow days in most areas exceeds 150 days, and in low altitude areas, or further south, SDD is less than ten days (Sun et al., 2020; Wang et al., 2018d). However, due to the existence of snow cover in the Asian mountains between 20° and 40° N, SDD varies greatly with altitude (Zhang and Ma, 2018). For example, SDD in the Himalayas is much longer than that in the
Qaidam Basin and the northern Tibetan Plateau (Zhang et al., 2016; Wang et al., 2017). Moreover, affected by the terrain, the snow in the northern Alps starts earlier, ends later and covers more days than that in the south under the same altitude conditions (Xie et al., 2017).

**3.2 Snow phenology variation and trend**

Fig. 2 exhibits the annual variation in snow phenology parameters over the Northern Hemisphere from 1972 to 2020 based
on NHSCE data. It is noted that there is a significantly advanced trend in the SCED, and SCOD has been occurring earlier, but the trend is not significant. Snow phenology has a relatively large interannual variability, and the trend of snow





phenology over a short period is not completely consistent with that over a long period, and is even contradictory in some instances. For example, in the past 50 year, SCD shows a decreasing trend but it has an increasing trend during 1982-2020 snow year.

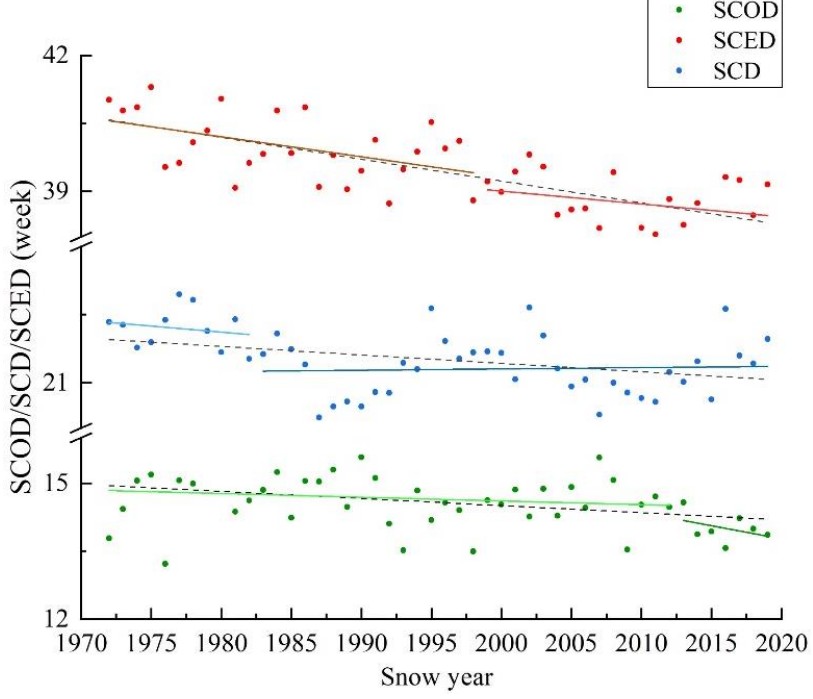

**Figure 2: Annual variation in SCD/SCOD/SCED from 1972 to 2020 based on NHSCE snow cover data.**

A multi-data set reveals that the spring snow melted in advance at a rate of 3-5 days per decade (d/10a), approximately 9-15 days earlier from 1970 to 2000 (Dye, 2002; Chen et al., 2015). The SCOD had no significant change (Callaghan et al., 2011; Chen et al., 2016), which is consistent with Fig. 2. Compared with 2001-2005, the onset dates and end dates of snow cover in the last five years (2015 to 2020) are 1.69 ± 5.93 days later and 0.94 ± 5.90 days earlier, respectively (Chen et al., 2021). However, it appears that in relatively low altitude plain areas, such as the West Siberian Plain and the Eastern European Plain, the SCOD tends to advance and the SCED is delayed, but the change is not significant (Sun et al., 2020).

The changing trends of snow phenology are very sensitive to the period of study. The study also found that the changes in snow phenology in different regions are not completely synchronous, and their contributions to the snow phenology variation in the whole Northern Hemisphere are quite different. Eurasia contributes more to the snow phenology variation in the Northern Hemisphere than does North America (Chen et al., 2016). The average SCODs in Eurasia and North America are both in late October, and the SCEDs are in early April and mid-April, respectively. Statistics based on ground observation data show that from 1982 to 2013, the SCED rapidly advanced at a rate of -2.6±5.6 d/10a in Eurasia, while it changed little in North America. The SCODs are delayed at rates of 1.1±4.9 d/10a and 1.3±4.9 d/10a in North America and



Eurasia, respectively (Peng et al., 2013). However, the changes in SCD have been almost synchronous in the pan Arctic regions of North America and Eurasia since 1966 (Callaghan et al., 2011).

For seasonal snow cover, the change in SCD is usually related to the delay or advance of SCOD and SCED (Liu et al., 2020). Many studies have shown that SDD has a shortening trend in the Northern Hemisphere (Zhang and Ma, 2018; Zhong et al., 2021, Peng et al., 2013), among which the SDD decreased by 0.8 week/10a between 1972 and 2008, and this change is

mainly caused by the advance of the SCED (Choi et al., 2010). In a study of snow cover in Canada, it was found that the shortening of the snow season is mainly due to the delay of SCOD (Brown et al., 2021). In addition, research based on MODIS data shows that the areas with significantly reduced SCD are primarily distributed in mountainous regions (Wang et al., 2018d).

## 4 Relationship between snow phenology and climate and vegetation

In the context of global warming, snow phenology has changed significantly, and the change in snow cover affects vegetation phenology. As shown in Fig. 3, there is a certain interaction between climate, atmospheric circulation, snow phenology and vegetation (Zhao et al., 2021; Huang et al., 2021; Rixen et al., 2001).

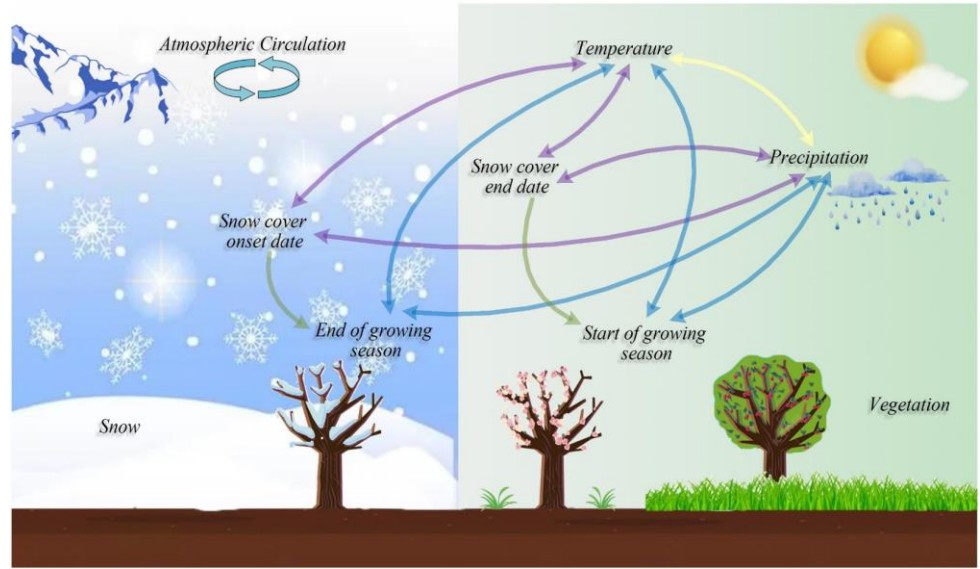

**Figure 3: The relationship between climate, atmosphere circulation, snow cover and vegetation.**

### 4.1 Relationship between snow phenology and climate

Seasonal snow cover is very sensitive to changes in meteorological factors, and there is a close relationship between snow cover and climate (Betts et al., 2014; Dutra et al., 2011; Xu and Dirmeyer, 2011). Climate warming will cause a change in the water vapor transport mode in the atmosphere, which affects the precipitation mode (Zhong et al., 2020; Rajeevan et al., 1998; Diro and Sushama, 2018; Diro et al., 2017). The rise in temperature directly delays the SCOD and advances the SCED





and reduces the SCD (Chapin et al., 2005). It has been found that for spring temperature increases of 1 °C, the average SCED advances by 1.6±1.8 days (Peng et al., 2013). The snow cover extent in the Arctic has decreased by $7\times10^5$-$8\times10^5$ km$^2$ due to climate warming, and according to the correlation analysis, there is a strong correlation between the snow phenology variation and snow cover extent in most cases (Dye, 2002). As far as the whole Northern Hemisphere is concerned, the SCED is highly related to temperature and precipitation during the snow accumulation season and temperature in the snow

melting season (Chen et al., 2021). However, in different latitudes, it may be affected by cold currents and human activities, thus, the dominant factors are not completely consistent. For example, in the northeastern United States, SCD is highly correlated with snowfall and temperature but not with precipitation (Leathers and Luff, 1997). The SCOD is determined mainly by the surface land temperature in the snow accumulation season and is most sensitive at approximately 40 °N (Chen et al., 2015).

Not only is snow phenology very sensitive to climate change, but the change in snow phenology could impact the climate system. According to statistics, a SCED advance of 1 day may cause temperature increases of 0.077±0.067 °C for that month; despite the fact that snow has little effect on temperature in autumn (Peng et al., 2013). The response of snow cover to climate is not completely uniform across different regions due to the influence of local climate and terrain. The rate of temperature rise in Eurasia is greater than that in North America during the snowmelt season, and the temperature in the two

regions begins to rise at a similar rate during snow accumulation season (Peng et al., 2013).

The decrease in snow cover caused by the rise in temperature will lead to a decrease in the surface albedo. The more shortwave radiation absorbed by the surface will cause an increase in the daily maximum temperature, which brings about the further melting of snow (You et al., 2010; Che et al., 2019). The changes in snow phenology and surface albedo lead to changes in circulation in nearby areas and even the whole Northern Hemisphere (Khandekar, 1991). Studies have found that

anomalous snow cover variation may be a major forcing factor for changes in atmospheric circulation in the Northern Hemisphere (Zhang et al., 2018), and anomalous changes in winter snow cover may cause an anomaly of East Asian monsoon circulation over Eurasia (Chen and Sun, 2003; Chen et al., 2003). In North America, extensive snow cover will decrease the sea surface temperature over the western North Atlantic by causing anomalous cold air advections downstream of the snow-cooled region. In addition, snow cover decreases surface air temperature via local diabatic cooling (Li et al.,

2020). However, some analyses show that the atmosphere drives the change in snow cover rather than snow forcing the atmosphere (Henderson et al., 2018).

Fig. 4 shows the dominant factors affecting the snow phenology variations and their effects. The main modes of atmospheric variability in the Northern Hemisphere include the Pacific North American (PNA), North Atlantic Oscillation (NAO) and Arctic Oscillation (AO) (Barnston and Livezey, 1987; Wallace and Gutzler, 1981). On a small scale, the change in snow

cover is more affected by temperature and precipitation, but on a large scale, the impact of atmospheric circulation on snow cover is stronger (Yang et al., 2019). In recent research on the snow cover variation in Eurasia, it was found that the winter snow cover in the region is affected mainly by the AO rather than climate warming (Yeo et al., 2017). In contrast, the reason for the decrease in spring snow cover in the Arctic may be more affected by the change in Arctic temperature than the role of





the AO (Brown et al., 2010). The positive (negative) patterns of the PNA are associated with the decreased (increased) snow
cover over western North America, and in the eastern United States, the opposite pattern is observed. The positive (negative)
patterns of the NAO are related to decreased (increased) snow cover across eastern North America and western Eurasia
(Brown and Goodison, 1996; Gutzler and Rosen, 1992; Serreze et al., 1998). The breakpoint of snow cover days on the
Tibetan Plateau in the 1990s is essentially consistent with the change in the AO index (You et al., 2011). Analyses of
multiple datasets reveal that even the same climate model may show different effects in Eurasia and North America due to
the complex regional characteristics of snow cover changes (Brown and Robinson, 2011). The response of atmospheric
circulation to snow cover in Eurasia is mainly concentrated in the downstream, whereas the response is more evident in the
upstream in North America (Henderson et al., 2013).

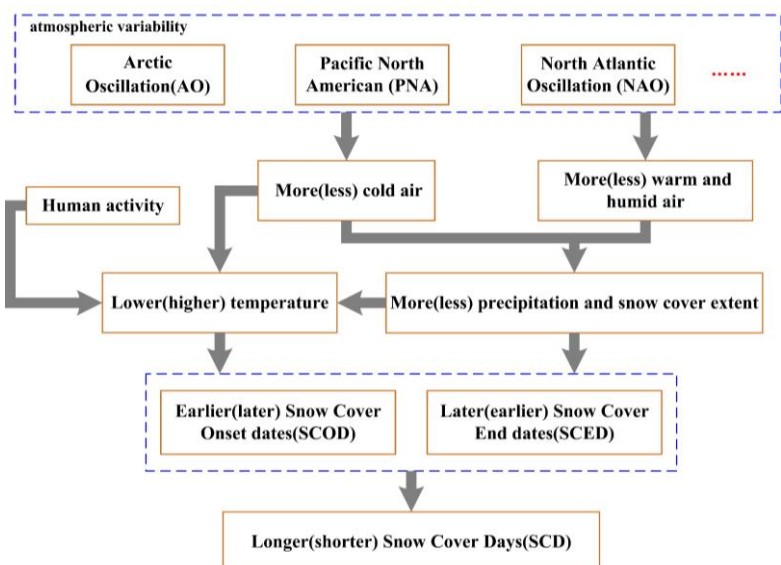

**Figure 4: Schematic of the dominant factors affecting snow phenology variations. This graphic is a summary of previous research**
**(You et al.,2020; Choi et al., 2010; Dye,2002; Brown et al.,2021; Chapin et al.,2005; Clark et al.,1999; Barnston and Livezey,1987;**
**Wallace and Gutzler,1981; Derksen et al., 2008)**

## 4.2 Relationship between snow phenology and vegetation

Vegetation phenology studies in snow covered forest regions are complex (Botta et al., 2000; Julien and Sobrino, 2009).
Vegetation can intercept the snow. The snow intercepted on the canopy can affect the energy exchange between the snow
and the ground, while the snow under the canopy is protected from direct solar radiation due to the shielding effect of the
canopy, which can delay the melting of the snow (Wang et al., 2013; Che et al., 2008). The change rate of snow phenology is
also significantly different under different types of vegetation (Qiao and Wang, 2019). The forest environment is relatively
closed, and the process of snow accumulation and melting is obviously different from that of nonforest areas. At the same
latitude, due to the conditions of forest areas themselves, there are more SCD in forest areas than in nonforest areas, and the
snow melting time occurs later (Guo et al., 2022; Gelfan et al., 2004).



Not only does vegetation affect the snow phenology variation, but snow cover also has a reaction to vegetation growth. Historically, snow cover has been considered an important factor affecting vegetation growth in alpine areas. Snow can provide water for vegetation growth, and its thermal insulation can also protect vegetation from wind and low temperatures (Paudel and Andersen, 2013). The start of the growing season is clearly impacted by advances or delays in snowmelt (Wipf

and Rixen, 2010). It is generally believed that in vegetation areas covered with snow, since snow will inhibit effective vegetation photosynthetic processes by blocking incoming radiation, the delay of SCED will postpone the start of, and shorten the length of the growing season; additionally, more snowfall will also do the same (Buus et al., 2006; Wang et al., 2015; Wang et al., 2018c; Jonas et al., 2008).

The response of vegetation phenology to snow phenology is different in different geographical regions and temperature and

precipitation gradients. The start of the spring growing season is approximately 1-5 weeks later than the SCED, which mainly depends on the air temperature after snow melting. The vegetation turns green earlier (later) when the temperature is higher (lower) than the average temperature after the SCED (Dong et al., 2013). However, there will be a negative correlation between the SCED and the start of vegetation growth in areas with relatively low precipitation. It may be that earlier snowmelt increases frost damage to plant buds (Inouye, 2008). In relatively humid areas, the longer SCD in previous

autumn and early winter promotes the vegetation growth. In arid areas, due to the shortening of SCD in non growing season, the start of spring vegetation growth is delayed (Huang et al., 2019). For example, snowmelt mainly promotes vegetation growth in the Arctic, northeast Eurasia and North America, while in Alaska and central Europe, snow primarily inhibits vegetation growth because more snow is usually expected to delay the SCED (Wang et al., 2018b). Obviously, there is a relationship between snow phenology and vegetation, and that relationship depends on elevation. It is found that in the Alps,

the correlation between snow cover and vegetation growth is significant at elevations of 1000-2000 m compared with other elevations (Asam et al., 2018).

## 5 Prospects

Snow phenology, a vital snow cover parameter, plays an important role in global snow cover change and climate change. At present, there are many datasets used in snow phenology research, but the characteristics of snow phenology obtained from

different datasets may differ. On the one hand, this is due to the inconsistently defined methods used in snow phenology research, often involving various observation data and the differing start times of the hydrological year, which can lead to considerable uncertainty in snow phenology results. On the other hand, the underlying principles regarding how snow information is obtained by the different observation methods are inconsistent, which also impacts the binary snow cover results. Furthermore, because the time span is different in different studies, the change characteristics of snow phenology are

not completely consistent even within identical regions (Zhong et al., 2021).



### 5.1 Analysis of snow phenology with multiple datasets

At present, there are still many uncertainties in the study of snow phenology. The accuracy and variable characteristics of the snow phenology parameters obtained by different sensors and different methods are not consistent. For example, when analyzing snow phenology variation during the snow season of full snow season (FSS) and continuous snow season (CSS),

the SCOD in FSS is basically unchanged, while it is significantly advanced in CSS (Choi et al., 2010). Therefore, determining how best to compare the consistency of data with different spatiotemporal resolutions, as well as time series and data products in different regions is very important for a comprehensive understanding and accurate grasp of snow phenology variation. The organic integration of ground observation data and different remote sensing monitoring data to form a set of more comprehensive and accurate snow cover products with high spatiotemporal resolution is an urgent

problem to be solved. In addition, we should make full use of the longer time scales and higher resolutions offered by reanalysis data to supplement existing data or as a source of high precision to extract snow phenology information.

### 5.2 Future changes in snow phenology

At present, research on snow phenology generally focuses on the spatiotemporal distribution of snow phenology characteristics and the factors influencing snow cover over the historical periods. There are a few studies pertaining to the

prediction of snow phenology variation using global climate models, and those that do mainly on estimating snow water equivalent and simulating snow cover extent (Roeach, 2006; Xia and Wang, 2015; Shi and Wang, 2015). Predicting changes in snow phenology is an important aspect in understanding cryosphere. Through reanalysis and environmental prediction models, the formation and melting of snow and the change characteristics of snow under different emission scenarios in the future can be simulated, and the relationship between snow phenology variation and climate can be better evaluated. This

can ultimately provide a scientific basis for the prediction of large scale water resource changes, climate change and even the prevention of extreme disasters in the future.

### 5.3 Physical mechanism of snow phenology variation

Relative to the study of snow cover extent, there are few studies about snow phenology in the Northern Hemisphere, and those that have been done mainly focus on a specific region (such as the Tibetan Plateau and Eurasia). At present, some

scholars have analyzed the interaction between snow cover extent and snow depth over a variety of climatic phenomena (Popova, 2007; Yeo et al., 2017; Chen and Wu, 2000), but there is a lack of research on the relationship between atmospheric circulation and climate change and snow phenology. The response mechanism between snow phenology variation and global climate change cannot be clarified in the Northern Hemisphere, and the reasons for the differences between snow phenology variation in different regions have not been clearly analyzed. It is proposed that this uncertain

reasons for the decline in snow cover will increase the uncertainty of predicting changes to future snow cover extents (Guo et





al., 2021). Similarly, the unclear physical mechanism in snow phenology will impact adequately predict its future changes and the related consequences.

### 5.4 Effects of human activities on snow phenology

Under global warming and due to the influence of human activities, snow cover will change significantly in the Northern
Hemisphere. In turn, changes in snow phenology will also bring certain changes to human society and the whole ecosystem. Snow phenology variation is very complex and is the result of the combined action of many factors, including natural factors and human activities. Some researchers infer that human caused climate warming is the leading factor for snow cover extent reduction in the recent (Derksen and Brown, 2012; Najafi et al., 2016; Willibald et al., 2020). Therefore, it is necessary to analyze the relative contributions of the various factors that influence changes to snow phenology. In addition, it is also
important to further quantitatively analyze the impact of snow phenology on various factors.

*Data availability.* The NHSCE dataset is provided by the National Oceanic and Atmospheric Administration (NOAA), which is openly accessible at https://nsidc.org/data/nsidc-0046/versions/4 (last access: 10 December 2022). The data used to obtain Figures 1 and 2 in a data repository is freely accessible at https://doi.org/10.5281/zenodo.7432273.

*Author contributions.* GH, WXY and GZC designed the study, performed the data analysis and wrote the manuscript; ZGF and CT designed the study; WJ and HXD inspected and revised the manuscript; HC and OYZQ downloaded and performed the data used in Fig.1 and Fig.2.

*Competing interests.* The authors declare that they have no conflict of interest.

*Acknowledgements.* We thank NASA official website for obtaining snow cover data (NHSCE), and we would also like to thank the anonymous reviewers for their valuable comments and thoughtful suggestions.

*Financial support.* This research has been supported by the National Natural Science Foundation of China (42125604, 42271373, 41971293).

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
