# Peer review of "Review of snow phenology variation in the Northern Hemisphere and its relationship with climate and vegetation"

_The Cryosphere, 2022_

## Referee Comment (RC1)

**Review of snow phenology variation**

By Hui Guo et al.

Even if it is a "Review article", it must make a significant contribution to the field, especially in a specialized journal like TC. This is not the case with this article. There is no structured, any classified synthesis and a lot of verbose!
The parameters used to describe snow (phenology: beginning, end and duration of snow cover) have a great spatial variability, and for each major region (North America (NA), Eurasia, Arctic) a specific evolution.
As the authors know, snow is the result of 2 phenomena: temperature and solid precipitations, 2 independent mechanisms (phenomena) evolving differently that are sometimes also interconnected but not always (it rains in winter!). It is well known, but here, it goes in all directions, with lists of cases referenced. Typically in Fig. 4, there are (more or less) in each box of the flowchart : so what? In Fig.3, precipitations:  liquid or solid?

Section 2: good Tables 1 and 2. But for the sections 2.1.3 and 2.1.4, recent references are missing (e.g. Mudryk et al.).

Section 3 should be presented as an example of the spatial variation of snow phenology averaged over 50 years (1970-2020) (Fig.1) and its temporal variation averaged over the whole Northern Hemisphere (NH) (Fig.2). OK, but so what?
In Fig.1, we can see the averaged latitudinal and altitudinal distribution. In the temporal variation (Fig.2), you have cut the trends in several sections, but over not significant periods! Moreover, when we know that the trend in NA is the opposite of that in Eurasia, what sense can be given to the global trend? Nevertheless, this figure is still interesting (with the global trend line over the whole period) and presents a global synthesis of the situation.

Sections 4 remains in generalities, without synthesis. The Fig.3 is simplistic and explains nothing.
The links with the vegetation are poorly explained. At the scale analyzed we talk about large biome, forest/non forest? This section is weak.

Section 5 Prospects: I do not agree that it has inconsistency on the detection methods of phenology, the simplest parameters to measure (done since 1966!). This may be true for other parameters such as depth, density or Snow Water Equivalent, but not for the onset, end and duration of snow cover. Depending of the scale of the analysis, the spatial resolution could be an issue. OK.
I don't understand the "human activity", you mean global warming generated by human activity?

In conclusion I do not recommend the publication of this article in TC. Perhaps, if restructured, publishable in a more generalist journal

---

## Author Comment (AC1)

Both reviews indicate that there may be some interest in your paper, however, they both note significant weaknesses. One important comment by Referee 2 is that you present a thorough analysis of data from the Tibetan plateau but that the data on other parts of the world is less convincing. Please improve your description and validation of general Northern hemisphere data, or perhaps, as suggested by the Referee, refocus your work on the Tibetan plateau only. That may also be a valuable contribution.

**Response:** As mentioned in our response to Referee 2, throughout the article, the data and analysis of snow phenology we presented are based on the Northern Hemisphere, not the Tibetan Plateau. In section 3, we display the distribution of snow phenology over the Northern Hemisphere from 1972 to 2020 based on NHSCE snow cover data, interannual variations over NA/EU/NH and the corresponding contributions to NH from EU and NA (NA, EU and NH represent Eurasia, North America and the Northern Hemisphere, respectively) (revised version). In section 4, we also focused on analyzing the interrelationships between snow phenology and temperature, atmospheric circulation, and vegetation at the scale of the Northern Hemisphere. For example, we mentioned in section 4.1 (revised version) that 'Studies have found that anomalous changes in winter snow cover may cause an anomaly in East Asian monsoon circulation over Eurasia (Chen and Sun, 2003; Chen et al., 2003), and spring Eurasian snow will affect the Indian summer monsoon through land–sea thermal differences and atmospheric circulation (Halder and Dirmeyer, 2017)'.

I agree with the Referees that the English is not satisfactory. Major improvements on this aspect are absolutely required.

**Response:** Thank you for your comments. We will have the paper polished by a native English speaking professional editor.

Referee 1 is fairly critical of your work. The paper may need significant restructuring. Sections that no not reach a useful and clear conclusion may have to be deleted. I am not convinced about the usefulness of Figures 3 and 4. In summary, very major changes to your paper are required before

serious consideration for publication can be made. The paper definitely needs to be condensed and focused on solid data. Unnecessary speculation or considerations should be deleted. I let you decide whether you can strengthen the paper and keep presenting data for the whole northern hemisphere or just focus on the Tibetan plateau. If you choose to consider the entire northern hemisphere, please explain in your response to Referee 2 how you have improved your analysis of northern hemisphere data in general.

**Response:** For section 4, we have restructured and supplemented it. Specific modifications have been shown in the response to Referee 1.

After sorting out section 4, we found that Fig. 4 is not completely suitable for the content of each paragraph in section 4.1, so we deleted it. Fig. 3 is a brief description of the interaction of snow phenology, vegetation phenology and climate factors. The purpose of this figure is to visually illustrate that there is a certain interaction between snow phenology, vegetation phenology and climate. Then, we separately introduce the relationship between snow phenology and climate and the relationship between snow phenology and vegetation phenology through two subsections.

Throughout this article, we have been devoted to the relevant analysis of snow phenology in the Northern Hemisphere. We have mentioned this in detail in our response to Referee 2.

Lastly, I think a few clarifications pertaining to Figure 1 are required. Please define clearly D1 and D2 in text, not as a Table footnote.

**Response:** Based on your opinion, we have added an explanation for D1 and D2 in the text rather than in the footnote.

Figure 1. How about having dates instead of weeks from 1 August? This would facilitate visualization. For Fig 1a, the scale would then be 1 September to 16 March. By the way, does a SCOD on 16 March make much sense? In Figure 1b, it is not clear to me what the dates are. Is week 27 from 1 August February 6th? This does not read very well. Reader should be able to figure out dates without having to compute them. Perhaps also add an intermediate date in the color scales for improved legibility.

**Response:** According to your opinion, we changed the legend to the form of date and added intermediate date, as shown in the figure below.

[Figure]

**Figure 1: Distribution of snow phenology over the Northern Hemisphere from 1972 to 2020 based on NHSCE snow cover data. Averaged over 48 snow years: (a) SCOD, (b) SCED, and (c) SCD (the study area excludes Greenland).**

---

## Author Comment (AC2)

Even if it is a "Review article", it must make a significant contribution to the field, especially in a specialized journal like TC. This is not the case with this article. There is no structured, any classified synthesis and a lot of verbose! The parameters used to describe snow (phenology: beginning, end and duration of snow cover) have a great spatial variability, and for each major region (North America (NA), Eurasia, Arctic) a specific evolution.

As the authors know, snow is the result of 2 phenomena: temperature and solid precipitations, 2 independent mechanisms (phenomena) evolving differently that are sometimes also interconnected but not always (it rains in winter!). It is well known, but here, it goes in all directions, with lists of cases referenced. Typically in Fig. 4, there are (more or less) in each box of the flowchart: so what? In Fig.3, precipitations: liquid or solid?

**Response:** Section 4 has been reorganized and supplemented. In section 4.1, each paragraph introduces the effect of climate on snow phenology, the response of snow anomalies to climate, the influence of snow cover on atmospheric circulation and the impact of circulation on snow cover. After sorting out section 4, we found that Fig. 4 was not completely suitable for the content of each paragraph in section 4.1, so we deleted it.

Fig. 3 provides a brief description of the interaction of snow phenology, vegetation phenology and climate factors. In the figure, precipitation refers to rainfall, so it is liquid.

**Two main modified paragraphs now read as follows:**

The decrease in snow cover caused by the rise in temperature will lead to a decrease in the surface albedo. More shortwave radiation absorbed by the surface will cause an increase in the daily maximum temperature, which results in further snow melting (You et al., 2010; Che et al., 2019). According to statistics, an SCED advance of 1 day may cause temperature increases by $0.077\pm0.067\,°C$ for that month (Peng et al., 2013), and snow cover will reduce the temperature of the lower troposphere locally by several degrees over days to months (Dewey, 1977). In North America, extensive snow cover decreases the sea surface temperature over the western North Atlantic by

causing anomalous cold air advections downstream of the snow-cooled region. In addition, snow cover decreases surface air temperature via local diabatic cooling (Li et al., 2020; Henderson et al., 2013). The response of snow cover to climate is not completely uniform across different regions due to the influence of local climate and terrain. The rate of temperature rise in Eurasia is greater than that in North America during the snowmelt season, and the temperature in the two regions begins to rise at a similar rate during the snow accumulation season (Peng et al., 2013). Based on the relationship between snow cover and temperature, we can make predictions (Li et al., 2020).

In recent years, the influence of snow cover on atmospheric circulation has attracted increasing attention. Extensive snow cover anomalies could lead to circulation variations in nearby areas and even throughout the Northern Hemisphere through energy budget changes in the lower atmosphere (Khandekar, 1991, Gong et al., 2003; Saito and Cohen 2003; Saunders et al., 2003; Zhang et al., 2018). Snow cover anomalies will change the stratospheric polar vortex through planetary waves, and then, its anomalous intensity will propagate downward, thus affecting the surface annular mode (Gong et al., 2003; Cohen and Saito, 2003). Studies have found that anomalous changes in winter snow cover may cause an anomaly in East Asian monsoon circulation over Eurasia (Chen and Sun, 2003; Chen et al., 2003), and spring Eurasian snow will affect the Indian summer monsoon through land−sea thermal differences and atmospheric circulation (Halder and Dirmeyer, 2017). When changes in snow cover over western and eastern Eurasia are positive and negative, respectively, the summer rainfall over East Asia could be enhanced (Yim et al., 2010). Too much snow in Eurasia leads to abnormal cooling of the overlying atmosphere. If a cold anomaly is established over the tropical Atlantic, it will lead to weak summer monsoon rainfall (Prabhu et al., 2017). However, some analyses show that the atmosphere drives the change in snow cover rather than snow forcing the atmosphere (Henderson et al., 2018), and changes in snow cover are the consequence of temperature and/or large-scale circulation variations (Derksen et al. 1997; Clark and Serreze, 2000).

The two paragraphs describe the impact of snow cover anomalies on climate and circulation patterns. The revised content about the impact of climate and atmospheric circulation on snow phenology is not shown.

Section 2: good Tables 1 and 2. But for the sections 2.1.3 and 2.1.4, recent references are missing

(e.g. Mudryk et al.).

**Response:** Thank you very much. Mudryk et al. (2020) presented an analysis of observed and simulated historical snow cover extent and snow mass, along with future snow cover projections from CMIP6models. It is an important work about Northern Hemisphere snow cover, and we have added this reference (Mudryk et al., 2020) to the paper.

Section 3 should be presented as an example of the spatial variation of snow phenology averaged over 50 years (1970-2020) (Fig.1) and its temporal variation averaged over the whole Northern Hemisphere (NH) (Fig.2). OK, but so what?

In Fig.1, we can see the averaged latitudinal and altitudinal distribution. In the temporal variation (Fig.2), you have cut the trends in several sections, but over not significant periods! Moreover, when we know that the trend in NA is the opposite of that in Eurasia, what sense can be given to the global trend? Nevertheless, this figure is still interesting (with the global trend line over the whole period) and presents a global synthesis of the situation.

**Response:** Fig. 1 presents the spatial variation in snow phenology averaged over 50 years (1970-2020) and shows that snow phenology has zonal and vertical distribution characteristics. In the original version, Fig. 2 presents the temporal variation, and we cut the period according to the snow phenological mutational year detected by the Mann-Kendall method. Based on your comments, we added some new analysis results and redrew Fig. 2. In the new Fig. 2, the trend in NA is consistent with that in Eurasia (Fig. 2a, b, c), and the conclusion is the same as that of Chen et al. (2016). Eurasia (EU) and North America (NA) are the two major regions in the Northern Hemisphere (NH). Through the contribution analysis method (Pederson et al., 2013), we counted the contribution of EU and NA to the NH trend (Fig. 2d, e, f).

[Figure]

**Figure 2: Interannual variations in (a) SCOD, (b) SCED, and (c) SCD over NA/EU/NH from 1972 to 2020 based on NHSCE snow cover data and the corresponding contributions to NH from EU and NA (NA, EU and NH represent Eurasia, North America and the Northern Hemisphere, respectively).**

**Our revised and supplemented paragraphs read as follows:**

Fig. 2a, b, c exhibits the annual variation in snow phenology parameters from 1972 to 2020 based on NHSCE data over the Northern Hemisphere (NH)/North America (NA)/Eurasia (EU). Eurasia and North America are the two major regions in the Northern Hemisphere, and their snow phenology changes are synchronous. There was an advanced trend in the SCED, and SCD slowly shortened in the past 50 years. Under the global warming trend, the SCODs in the EU, NA and NH have shown early trends. This conclusion seems to be incorrect, but a new study explains the rationality of this phenomenon. A clear relationship between earlier (and later) onset trend patterns was found at 500 hPa geopotential heights and sea level pressure, airflows at 500 and 850 hPa, atmospheric humidity, and near-surface temperature (Allchin and Déry, 2020). Another interesting conclusion is that the snow cover in North America starts earlier and ends later than that in Eurasia. We infer that due to high regional variability, it may be driven by different regional warming, precipitation rates and atmospheric circulation patterns, as mentioned above.

The study also found that the changes in snow phenology in Eurasia and North America have different contributions to the whole Northern Hemisphere (Fig. 2d, e, f). According to the contribution analysis method (Pederson et al., 2013), Eurasia contributes more to snow phenology

variation in the Northern Hemisphere than does North America, which is consistent with previous studies (Chen et al., 2016). The change rate of snow phenology in Eurasia is closer to that in the Northern Hemisphere (Fig. 2a, b, c). Statistics show that the SCED change rate in Eurasia is faster than that in North America, while the SCOD change rate in Eurasia is slower than that in North America in the past 50 years. The SCDs have shortened at a rate of -0.2 d/10a in Eurasia, while SCD changed little in North America. However, the changes in SCD have been almost synchronous in the pan-Arctic regions of North America and Eurasia since 1966 (Callaghan et al., 2011).

Sections 4 remains in generalities, without synthesis. The Fig.3 is simplistic and explains nothing. The links with the vegetation are poorly explained. At the scale analyzed we talk about large biome, forest/non forest? This section is weak.

**Response:** Thank you for pointing out the problem in section 4. In the new manuscript, section 4 has been reorganized and supplemented according to your comments.

(1) Fig. 3 provides a brief description of the interaction among snow phenology, vegetation phenology and climate factors. In the previous manuscript, the necessary explanation of Figure 3 was missing.

**We have added some description about Figure 3 as follows:**

Snow cover is closely coupled to the soil-climate-vegetation system with multiple interactions and feedbacks. In the context of global warming, snow phenology has changed significantly, and changes in seasonal snow cover regulate regional and global climatic systems and vegetation growth by changing the energy budgets of the atmosphere and land surface (Yu et al., 2013). Similarly, changes in vegetation and snow cover may cause climate change through surface albedo and energy exchange between land and atmosphere (Euskirchen et al., 2016). As shown in Fig. 3, there is a certain interaction among climate, atmospheric circulation, snow phenology and vegetation (Zhao et al., 2021; Huang et al., 2021; Rixen et al., 2001). The start of the growing season (SOS) is clearly impacted by advances or delays in snowmelt (Wipf and Rixen, 2001). The snow-vegetation interaction has a positive feedback effect on warming at northern latitudes (Royer et al., 2021). A study suggests that in mountainous areas, snow cover and temperature can be combined to produce a more applicable and accurate vegetation phenology model (Asam et al., 2018).

(2) The content of the relationship between snow phenology and climate has also been supplemented and has been discussed in the first question.

(3) Section 4.2 summarizes the relationship between snow phenology and vegetation. The main purpose is to summarize the snow cover-vegetation interactions in the vegetation area covered by snow, and the vegetation area includes tundra, spruce and shrub.

In this section, we describe the impact of vegetation on snow cover, the impact of snow cover on vegetation growth, and the relationship between snow phenology and vegetation phenology under different conditions.

**Each paragraph is supplemented as follows:**

In dense high shrubs with stems above the snow surface, the longwave radiation may increase, thus accelerating snow melting. In open forest areas, the melt rate appears to be slightly lower, possibly due to shadowing effects (Marsh et al., 2010). Similarly, low-lying plants covered by snow have a higher albedo in winter, while tall plants with exposed canopy have a lower albedo (Sturm et al., 2005). When spring arrives, the low-lying plant area can absorb more solar radiation, resulting in earlier SCOD and SOS.

Some studies have even shown that changing snow conditions lead to changes in plant coverage (Sandvik and Odland, 2014). It is generally believed that in vegetation areas covered with snow, since snow will inhibit effective vegetation photosynthetic processes by blocking incoming radiation, snow phenology will influence the start and length of vegetation phenology (Trujillo et al., 2012, Yu et al., 2013, Paudel and Andersen, 2013). For instance, the delay in SCED will postpone the start of and shorten the length of the growing season; additionally, more snowfall will have the same effect (Buus et al., 2006; Wang et al., 2015; Wang et al., 2018c; Jonas et al., 2008). In contrast, shorter SCD and earlier SCED often advance vegetation growth (Chen et al., 2011; Wipf et al., 2009; Wipf & Rixen, 2010). The SCD during nongrowing seasons may also show a lag effect on the SOS to a certain extent (Huang et al., 2019).

In addition, there is a relationship between snow phenology and vegetation, and the strength of the relationship is related to the latitude bands. In the Alps, the correlation between snow cover and vegetation growth changed significantly with altitude, and SCD played a key role in the SOS at middle and high altitudes (Xie et al., 2017). There is a delay of SOS by 2.5 days and a shortening of SCD by 10 days per 100 m. At low latitudes, it may be affected by human activities and deviate

from this trend (Asam et al., 2018). In addition, the vegetation phenology of various land cover features has a different response to snow phenology, and the lag effect of snow phenology variations on vegetation phenology mainly depends on vegetation type. There was a significant relationship between the SOS and SCED in the low shrubs, and the SCED occurred earlier than the SOS by less than five days. The end of the growing season (EOS) and SCOD have a strong relationship in almost every vegetation type. EOS was later than SCOD for almost all vegetation types except mountain vegetation (Zeng et al., 2013).

Section 5 Prospects: I do not agree that it has inconsistency on the detection methods of phenology, the simplest parameters to measure (done since 1966!). This may be true for other parameters such as depth, density or Snow Water Equivalent, but not for the onset, end and duration of snow cover. Depending of the scale of the analysis, the spatial resolution could be an issue. OK. I don't understand the "human activity", you mean global warming generated by human activity

**Response:** In section 2.2, we introduced three methods of defining snow phenology. Various definition methods will cause differences in the statistical results; this speculation is also mentioned in Zhong (2021). Taking the snow cover onset date (SCOD) as an example, if it is defined as the first day snow is monitored in a hydrological year, it may be affected by instantaneous snowfall or abnormal conditions (the pixel is incorrectly divided into snow). Therefore, we generally believe that the SCOD is the first day on which the pixel is first covered with snow for three or five consecutive days (temporarily recorded as D1). However, in the Qinghai-Tibet Plateau, the distribution of snow cover is unstable, and most areas are covered with patchy snow, with a short time sequence. In this case, using D1 to identify the SCOD will introduce some errors. Therefore, different definition methods will lead to inconsistent snow phenology results. Perhaps we did not express this clearly in the text, and we described it again in the revised manuscript.

**This sentence changed to:**

On the one hand, various snow phenology definition methods will cause differences in the statistical results.

In this article, human activity refers to anthropogenic climate change, including the atmospheric concentrations of greenhouse gases and aerosols (Najafi et al., 2016; Willibald et al.,

2020). To express this more clearly, we replaced human activity with anthropogenic forcing.

**Now the paragraph reads as follows:**

Snow phenology variation is very complex and is the result of the combined action of many factors, including natural climate variability and anthropogenic climate change. Although it is important to understand the impact of natural climate variability on snow phenology, the change in snow phenology is likely to change with anthropogenic forcing. Some researchers have inferred that anthropogenic climate warming has been an important factor in snow cover extent reduction in recent years (Derksen and Brown, 2012; Najafi et al., 2016; Willibald et al., 2020). The latest research found that climate warming caused by greenhouse gases is the main reason for the Northern Hemisphere snow cover extent reduction, but aerosols play a positive role in slowing snow cover extent decline, offsetting approximately 16% (Guo et al., 2021). However, it is unclear how anthropogenic forcing, natural variability and greenhouse gases affect snow phenology in the Northern Hemisphere. Therefore, it is necessary to analyze the relative contributions of the various factors that influence changes in snow phenology. In addition, it is also important to further quantitatively analyze the impact of snow phenology on various factors.

**Added references are as follows:**

Mudryk, L., Santolaria-Otín, M., Krinner, G., Ménégoz, M., Derksen, C., Brutel-Vuilmet, C., Brady, M., and Essery, R.: Historical Northern Hemisphere snow cover trends and projected changes in the CMIP6 multi-model ensemble, Cryosphere., 14, 2495–2514, doi: 10.5194/tc-14-2495-2020, 2020.

---

## Author Comment (AC3)

There is a lot of good information in this paper. The authors have done a good job of researching the literature, especially as it relates to snow cover in China. They provide a short but nice review of remote sensing data products in Section 2.1.2 and in Table 2. Additionally, they address an important topic that deals with cyclic change in the Northern Hemisphere snow cover patterns, i.e., snow phenology, as it is related to climate change.

In the early part of the Introduction the authors should define 'snow phenology' so that the reader understands the exact meaning with respect to this paper. It is defined in the first sentence of the Abstract, but needs to be defined more clearly using proper English there. The words 'snow phenology' are used repeatedly throughout the manuscript, thus the authors need to ensure that the phrase is referred to in a consistent manner throughout and matches the exact definition that they need to add to the Introduction.

**Response:** Thank you for your suggestion. We also intended to define snow phenology in the abstract or introduction, but we did not find a clear definition in the previous papers.

Chen et al. (2021) described the snow phenology—'Snow cover phenology (SCP) variables including snow onset date (Do), snow end date (De), and snow duration days (Dd) are key indicators of seasonal variation of terrestrial snow cover over the NH and becoming increasingly valuable indicators of climate change.'

Notarnicola et al. (2020) described snow 'Changes in snow cover and related phenology (duration, onset and melt) have a critical role in mountain environment, and are strictly related to water availability in downstream areas.'

We have checked and ensured that the snow phenology, snow cover onset date (SCOD), snow cover end date (SCED) and snow cover days (SCD) are referred to in a consistent manner throughout the article.

I am not clear about whether this review paper belongs in The Cryosphere. Perhaps the authors could re-focus the paper on snow cover on the Tibetan Plateau?? I don't think they have done a

thorough job addressing snow cover in the entire Northern Hemisphere.

**Response:** Throughout the article, the data and analysis of snow phenology we presented are based on the Northern Hemisphere, not the Tibetan Plateau. In section 3, we display the distribution of snow phenology over the Northern Hemisphere from 1972 to 2020 based on NHSCE snow cover data, interannual variations over NA/EU/NH and the corresponding contributions to NH from EU and NA (NA, EU and NH represent Eurasia, North America and the Northern Hemisphere, respectively). In section 4, we also focused on analyzing the interrelationships among snow phenology and temperature, atmospheric circulation, and vegetation at the scale of the Northern Hemisphere. For example, we mentioned in section 4.1 (revised version) that 'Studies have found that anomalous changes in winter snow cover may cause an anomaly in East Asian monsoon circulation over Eurasia (Chen and Sun, 2003; Chen et al., 2003), and spring Eurasian snow will affect the Indian summer monsoon through land–sea thermal differences and atmospheric circulation (Halder and Dirmeyer, 2017)'.

Do we really need all of those remote sensing products to address the snow phenology questions? It almost seems like there are two papers here – one that reviews remote sensing products and another that addresses the snow phenology question.

**Response:** At present, there are many data available for snow cover research, and we can use a multi-dataset approach to map snow cover over the NH because it exploits the strengths of the various platforms and methodologies. Analysis of multiple datasets or an ensemble product helps to overcome the limitations of individual datasets. Brown (2010) conducted a multi-data set analysis of variability and change in Arctic snow cover, examined the consistency of Arctic snow cover products, and provided estimates of the uncertainty in snow cover extent.

Therefore, it is important to be familiar with a variety of remote sensing products to address snow phenology questions. The snow cover product is the basis for obtaining snow phenology. Only by obtaining accurate snow phenology information can we correctly understand the relationship between climate change and snow.

Work is needed to improve the English in this paper. Overall it is acceptable but there are many places that need work. I only mentioned a few, below, though there are many other places that need

improvement in the English on every page.

**Response:** We will have the paper polished by a native English speaking professional editor.

The authors should consider removing the word 'variation' from the title, and perhaps replacing it with the word 'literature,' because you cannot review variation

**Response:** Thank you for your opinion. Indeed, 'variation' is not appropriate here.

**The title was changed as follows:**

Review of snow phenology in the Northern Hemisphere and its relationship with climate and vegetation

Specific comments:

Line

36 delete: "which significantly affects Earth's climate change."

**Response:** We have deleted this sentence.

54 the Warren references is in appropriate here since climate change was not really written about much in 1982.

**Response:** For the quotation of this paragraph---'snow phenology plays an important feedback role in climate change through its characteristics of high reflectivity and low thermal conductivity', we have added new references (Ke et al., 2016; Notarnicola et al., 2020), which describe snow phenology as playing an important role in climate change.

Table 1 and write-up preceding the table: what about snow depth datasets in the United States?

**Response:** The statistical content of Table 1 was referred from Peng et al. (2013), and the snow depth dataset in the United States can be freely obtained from GHCN. We can obtain the day, month, and year of interest for snow depth observation data from GHCN stations, which is accessible at **https://www.ncei.noaa.gov/access/monitoring/daily-snow/AK/snow-depth/ 20230201**.

102 Passive microwave products do not have very good accuracy for mapping snow, snow depth and SWE

**Response:** For the description of passive microwave data, we added the following sentence. The corresponding reference was added. In this reference, it described that due to topographic and land cover heterogeneity, spatial variability in SWE derived from passive microwave satellite measurements is not adequately captured. Coarse resolution is a particularly critical limitation in alpine regions, which are masked out completely in some products.

**Added the sentence as follows:**

however, it does not have good accuracy for mapping snow cover due to the rough resolution (Mortimer et al., 2020)

105 the word "influential" is not appropriate in this context

**Response:** This sentence was revised as follows:

Global Snow Monitoring for Climate Research (Globsnow), which was released by the European Space Agency in 1979, is a daily global SWE dataset with a spatial resolution of 25 km (Lin et al., 2020).

112 what is "L1R?"

**Response:** L1R means Level-1R. AMSR2 Level-1R (L1R) input brightness temperatures that are calibrated, or unified, across the JAXA AMSR-E and JAXA AMSR2 L1R products.

We went to the website to obtain the AMSR-E/AMSR2 and were surprised to find that the description of data is now 'This AMSR-E/AMSR2 Unified Level-3 (L3) dataset provides daily estimates of Snow Water Equivalent (SWE)'.

**we modified the original sentence as follows:**

The Advanced Microwave Scanning Radiometer2 (AMSR2) onboard the GCOM-W1 satellite was launched in 2012 as a follow-up product to AMSR-E, and the daily SWE data can be derived from AMSR2 L3 (Tedesco and Jeyaratnam, 2019).

Table 2: all of the MODIS products are global, including MOD10A1, MOD10A2, MOD10C2 and MOD10CM; also all of the MODIS products extend from 2000-present.

**Response:** On the MODIS website, we further confirmed that the MOD10A1, MOD10A2, MOD10C1, MOD10C2 and MOD10CM products are global and extend from 2000-present.

However, not all MODIS products start in 2000; for example, MYD10A1, MYD10A2, and MYD10C2, which were carried on the Aqua satellite, extend from 2002-present.

183 and 211 the word "plain" should read "plains"

**Response:** Thank you for your comments, and we have modified these words.

200 delete the word "significantly"

**Response:** Thank you for your comments, and we have deleted this word.

Figure 2: please provide a reference for the caption; where has this analysis been published?

**Response:** Figure 2 shows our own statistical results using NHSCE data, rather than referring to other papers.

241 over what time period has the snow cover extent in the Arctic decreased?

**Response:** My apologies, perhaps I did not express it clearly. What I meant to say was the impact of temperature rise on the Arctic snow cover extent, rather than describing a decrease in the Arctic snow cover extent over a certain period of time. We have rephrased the original text.

**The sentence now reads as follows:**

It has been found that when the temperature increases by 1 °C, the average SCED advances by 1.6±1.8 days (Peng et al., 2013), and the snow cover extent in the Arctic decreases by $7 \times 10^5$-$8 \times 10^5$ $km^2$ (Derksen and Brown, 2012).

333 delete the word "organic"

**Response:** Thank you for your comments, and we have deleted this word.

352-354 I am not quite sure what this sentence means??

**Response:** We apologize that we did not describe the meaning of this sentence clearly.

**We have rephrased this sentence to read:**

The response mechanism between snow phenology variation and global climate change is not clarified in the Northern Hemisphere, and the reasons for regional differences in snow phenology

variation have not been clearly analyzed. For example, Figure 2b shows that the SCED in North America is later than that in Eurasia. We speculated that this phenomenon might be related to the atmospheric circulation pattern according to previous studies, but there is no relevant research evidence.

**Added references are as follows:**

Ke, C.Q., Li, X.C., Xie, H., Ma, D.H., Liu, X and Kou, C.: Variability in snow cover phenology in China from 1952 to 2010, Hydrol. Earth. Syst. Sci., 20:755–770, doi:10.5194/hess-20-755-2016, 2016.

Mortimer, C., Mudryk, L., Derksen, C., Luojus, K., Brown, R., Kelly, R and Tedesco, M.: Evaluation of long-term Northern Hemisphere snow water equivalent products, Cryosphere., 14, 1579–1594, doi: 10.5194/tc-14-1579-2020, 2020.

Notarnicola, C.: Hotspots of snow cover changes in global mountain regions over 2000–2018, Remote. Sens. Environ., 243, 111781. doi: 10.1016/j.rse.2020.111781, 2020.

Tedesco, M. and J. Jeyaratnam. (2019). AMSR-E/AMSR2 Unified L3 Global Daily 25 km EASE-Grid Snow Water Equivalent, Version 1 [Data Set]. Boulder, Colorado USA. NASA National Snow and Ice Data Center Distributed Active Archive Center. https://doi.org/10.5067/8AE2ILXB5SM6.